# Long term risk and costs of bleeding in men and women treated with triple antithrombotic therapy–An observational study

**Anna Holm[1], Martin Henriksson[2], Joakim Alfredsson[1], Magnus Janzon[1], Therese Johansson[2], Eva Swahn[1]\*, Dominique Vial[2], Sofia Sederholm Lawesson[1]**

**1** Department of Cardiology and Department of Health, Medicine and Caring Sciences, Linköping University, Linköping University Hospital, Linköping, Sweden, **2** Department of Health, Medicine and Caring Sciences, Linköping University, Linköping, Sweden

\* eva.swahn@liu.se

**Data Availability Statement:** All relevant data are within the manuscript and its Supporting Information files.

## Abstract

### Objectives

Bleeding is the most common non-ischemic complication in patients with coronary revascularisation procedures, associated with prolonged hospitalisation and increased mortality. Many factors predispose for bleeds in these patients, among those sex. Anyhow, few studies have characterised the population receiving triple antithrombotic therapy (TAT) as well as long term bleeds from a sex perspective. We investigated the one year rate of bleeds in patients receiving TAT, potential sex disparities and premature discontinuation of TAT. We also assessed health care costs in bleeders vs non-bleeders.

### Setting

Three hospitals in the County of Östergötland, Sweden during 2009–2015.

### Participants

All patients discharged with TAT registered in the SWEDEHEART registry.

### Primary and secondary outcome measures

All bleeds receiving medical attention during one-year follow-up were collected by retrieving relevant information about each patient from medical records. Resource use associated with bleeds was assigned unit cost to estimate the health care costs associated with bleeding episodes.

### Results

Among 272 patients, 156 bleeds occurred post-discharge, of which 28.8% were gastrointestinal. In total 54.4% had at least one bleed during or after the index event and 40.1% bled post discharge of whom 28.7% experienced a TIMI major or minor bleeding. Women discontinued TAT prematurely more often than men (52.9 vs 36.1%, p = 0.01) and bled more (48.6

**Funding:** This study received funding from the County Council of Östergötland.

**Competing interests:** The authors have declared that no competing interests exist.

vs. 37.1%, p = 0.09). One-year mean health care costs were EUR 575 and EUR 5787 in non-bleeding and bleeding patients, respectively.

## Conclusion

The high bleeding incidence in patients with TAT, especially in women, is a cause of concern. There is a need for an adequately sized randomised, controlled trial to determine a safe but still effective treatment for these patients.

## Introduction

Bleeding is the most common non-ischemic complication in patients with myocardial infarction (MI), and has gained much attention during the last years due to its large impact on health outcomes [1–3]. Reduction of ischemic events with more potent antithrombotic therapies has come at the expense of more bleeding, which itself is associated with prolonged hospitalisation [2,4,5] and increased mortality [6]. Many factors predispose for bleeding, such as use of anticoagulants, age, renal failure and a history of previous bleeding [1,7–10]. Women bleed more than men when treated with dual antiplatelet therapy (DAPT) [8,9,11], even if the reduced risk for a new clot is equal [4,5,12].

DAPT is the corner-stone treatment post MI and after percutaneous coronary intervention (PCI), in order to prevent new coronary ischemic events, including stent thrombosis (ST) [13]. Approximately 10% of patients with acute coronary syndrome (ACS) require long-term oral anticoagulants (OAC) because of prosthetic heart valves, thromboembolism or atrial fibrillation (AF) [14]. OAC has been found superior to DAPT in order to prevent venous thromboembolic events (VTE) and stroke [15], and thus these patients are often discharged with both DAPT and OAC, so called triple antithrombotic therapy (TAT) [16]. Previous studies have shown that patients with TAT have up to four times higher risk of major bleeds than those with OAC [17], and twice the risk of haemorrhagic stroke compared with DAPT [16]. Recurrent cardiovascular events as well as bleeds are a burden not only for the patient but also for the health care sector and for the society at large [18].

Very few studies have characterised the whole TAT population as well as long term bleeding complications from a sex perspective. We hypothesised that women bleed more than men, and discontinue TAT more often.

## Aims

The primary aims were to investigate the rate of bleeding complications during the first year of follow-up in TAT treated patients, and study sex differences in rate of bleeds and premature discontinuation of TAT.

Secondary aims were to investigate bleeds during index hospitalisation and to explore resource use and healthcare costs in bleeders vs non-bleeders.

## Methods

### Study population and data collection

We identified patients discharged with TAT from three cardiology departments in the County of Östergötland, Sweden, between 1 January 2009 and 31 December 2015. Patients from other counties were excluded due to lack of access to their patient files during follow up. No other inclusion or exclusion criteria were applied. We used data from the Swedish Web-system for

Enhancement and Development of Evidence- based care in Heart disease Evaluated According to Recommended Therapies (SWEDEHEART) (http://www.ucr.uu.se/swedeheart) register which includes patients hospitalised because of suspected or diagnosed ACS. SWEDEHEART is a web-based national quality registry and data are registered online [19]. In the County of Östergötland approximately 800 patients with ACS are treated annually, and are registered in SWEDEHEART. The details of the register have previously been published [32]. For patients with more than one hospitalisation during the inclusion time window, only data from the first one was recorded. A predefined template was developed to retrieve relevant information about each patient from their medical records. The review of the patient records was performed by five of the authors (AH, ES, SSL. TJ and DV) and in case of uncertainty events were assessed once more in the group. Over 200 variables were extracted including comorbidities, bleeds, medication and laboratory data. Each bleeding complication receiving attention in the medical records was reported. From the registry 274 unique patients were identified, of whom two were excluded in the medical record retrieving data process, as they were never prescribed TAT. Our final study population consisted of 272 patients.

## Outcomes

The primary outcomes were the occurrence of any bleeding complication receiving medical attention and premature discontinuation of TAT within one year after discharge from the index event. Secondary outcomes were bleeding complications in-hospital, total rate of bleeding events, and health care costs per patient. Two standard bleeding classifications were used to measure the severity of each bleeding event, TIMI (Thrombolysis In Myocardial Infarction) [20], and BARC (Bleeding Academic research Consortium) [21].

Resource use data retrieved in the study included hospitalisations, outpatient care visits, procedures and blood products. Unit costs used to value resource use and calculate health care costs were derived from administrative data bases at Linkoping university hospital [22].

## Ethics

All patients were informed of their participation in the SWEDEHEART registry, and their possibility to withdraw their consent at any time. Anyhow, according to Swedish regulations, written informed consent is not required for registration in national quality registries such as SWEDEHEART. Permission for the study was obtained from the regional Ethical Review Board, Linkoping, Sweden (Dnr 2015/49-31; 2016/4040-32), and complied with the Declaration of Helsinki. Patient data were anonymised to protect integrity.

## Statistics

All continuous variables had a normal distribution of data and are thus presented by their mean and standard deviation (SD). Categorical variables are presented as counts and percentages. The Students´ T-test was used when comparing continuous variables with a normal distribution of data between men and women. When comparing the sexes regarding continuous variables with a non-normal distribution, the Mann Whitney U-test was performed. The Chi square test was used when comparing the distribution of categorical variables. When expected cell size was below five, Fisher's Exact test was used instead. The time to first bleeding receiving medical attention was estimated using Kaplan-Meier curves, and men and women compared using the log-rank test. A Cox proportional hazard regression analysis was conducted to find out factors independently associated with long term risk of bleeding. Factors included in the regression analysis test were chosen based on clinical and theoretical relevance; age, sex, laboratory measures on admission (haemoglobin and creatinine), previous comorbidities (stroke,

MI, bleeds), and OAC initiated before or during hospitalisation of the index event. Hazard ratios (HR) with 95% confidence intervals (CI) are presented for the variables significantly associated with long term bleeding. A p-value <0.05 is considered to indicate statistical significance.

Health care costs are summarised in terms of descriptive statistics and reported as per patient means and standard errors. Costs are reported by bleeding status during follow-up, and cost associated with bleeding with certainty is reported separately from those not necessarily associated with bleeding. Health care costs by sex were also explored.

Statistical analyses were performed with the IBM SPSS Statistics version 24 software and Stata version 14 (Stata Statistical Software: Release 7.0. College Station, TX, USA: Stata Corporation).

## Results

### Baseline characteristics

In this cohort of 272 patients (74.3% men, 25.7% women) the mean age was 74.1 years and women were older than men (77.4 vs 73.0 years, p<0.01). Almost half of the population (44.9%) had a history of ischemic heart disease, 34.9% had a previous MI, 31.6% had experienced a previous bleeding. Diabetes was more prevalent in women (38.6 vs 26.2%, p = 0.05). On admission 60.3% were treated with OAC, use of proton pump inhibitors (PPI) were more common in women (42.9 vs 28.7%, p = 0.03). Women had lower haemoglobin (129.4 vs 141.3 g/L, p<0.01) and creatinine (87.1 vs 103.3 μmol/L, p<0.01) on admission (Table 1).

### Indications for DAPT and OAC

The most common indications for DAPT upon index discharge were non-ST-elevation MI (NSTEMI) (38.6%) and ST-elevation MI (STEMI) (38.2%). The remaining patients had either unstable angina pectoris (UAP) (15.1%) or were hospitalised because of stable CAD (8.1%). STEMI was more common in men (41.6 vs 28.6%, p = 0.05) whereas NSTEMI was more common in women (48.6 vs 35.1%, p = 0.05). Indications for OAC were AF (71.3%), mechanical valve (3.7%), venous thromboembolism (11.8%), mural thrombus (7.0%), ventricular aneurysm (2.6%), stroke/transient ischemic attack (TIA, 2.2%) and coagulation disorders (1.5%). There were no sex differences in indications for OAC.

### Interventions and bleeding events in-hospital

Coronary angiography was performed in 97.4%, and 93.4% were treated with PCI. Radial artery was used as access site in 61.0% with no significant sex differences. Twenty-five percent of the patients had at least one bleeding complication during index hospitalisation of whom 55.9% were related to angiography or PCI. These bleeding episodes were classified as minimal according to TIMI bleeding scale (82.4%) or BARC 1 grade (61.8%). No significant sex differences in bleeding incidence, site or severity were found, except that women got blood transfusions more often (4.2% vs 1.0%, p = 0.02) (Table 2, S1 Table).

### Planned duration and premature discontinuation of TAT

Most patients (61.8%) were planned for ≤1 month of TAT followed by 11 months of aspirin and OAC combined, followed by OAC only. The use of direct acting OAC (DOAC) was low, 97.1% were discharged with warfarin. Clopidogrel was almost exclusively used as $P2Y_{12}$-inhibitor, with 7 cases (2.6%) using ticagrelor and none using prasugrel. In 27.6% TAT was planned for 1–3 months, in 9.9% 3–6 months, and 0.7% had 6–12 months, with no sex differences.

TAT was discontinued earlier than intended in 110 patients (40.4%), more in women (52.9 vs 36.1%, p = 0.01). In 44 patients (16.2%) aspirin was discontinued, significantly more in women (25.7 vs 12.9%, p = 0.01). In 31 patients (11.4%) the $P2Y_{12}$-inhibitor was discontinued, 17.1 and 9.4% in women and men respectively (p = 0.08). In 91 patients (33.5%) OAC was discontinued prematurely, without sex difference (37.1 vs 32.2%, p = 0.45). Bleeding was the most common reason why TAT was discontinued in both women and men, explaining 45.9 and 43.8% (p = 0.83) of discontinued TAT treatment, respectively (Table 3, S2 Table).

## Bleeding events during follow-up

A total of 40.1% (109) had at least one bleed. Seventy-eight patients had one bleed whereas the remaining 31 patients had up to 4 bleeds. Women had a nonsignificant higher rate of bleeds (48.6 vs 37.1%. p = 0.09). (Fig 1).

**Table 1. Baseline characteristics.**

| | All n = 272 | Women n = 70 | Men n = 202 | P-value |
|---|---|---|---|---|
| Age in years. mean (SD) | 74.1 (9.9) | 77.4 (9.6) | 73.0 (9.9) | <0.01 |
| **Risk factors and comorbidity** | | | | |
| Diabetes mellitus | 80 (29.4) | 27 (38.6) | 53 (26.2) | 0.05 |
| Ischemic stroke or TIA | 42 (15.4) | 10 (14.3) | 32 (15.8) | 0.76 |
| Renal insufficiency | 10 (3.7) | 4 (5.7) | 6 (3.0) | 0.29 |
| Ischemic heart disease | 122 (44.9) | 34 (48.6) | 88 (43.6) | 0.47 |
| Myocardial infarction | 95 (34.9) | 28 (40.0) | 67 (33.2) | 0.30 |
| Previous PCI | 67 (24.5) | 17 (24.3) | 50 (24.6) | 0.94 |
| Previous CABG | 47 (17.3) | 12 (17.1) | 35 (17.3) | 0.97 |
| Liver disease | 4 (1.5) | 2 (2.9) | 2 (1.0) | 0.27 |
| Peptic ulcer | 6 (2.2) | 1 (1.4) | 5 (2.5) | 1.00 |
| Bleeding | 86 (31.6) | 18 (25.7) | 68 (33.7) | 0.22 |
| Previous GI endoscopy* | 50 (18.4) | 15 (21.4) | 35 (17.3) | 0.45 |
| **Medication on admission**** | | | | |
| NSAID | 6 (2.21) | 2 (2.9) | 4 (2.0) | 0.65 |
| PPI | 88 (32.4) | 30 (42.9) | 58 (28.7) | 0.03 |
| Aspirin | 71 (26.1) | 22 (31.4) | 49 (24.3) | 0.24 |
| Clopidogrel | 11 (4) | 1 (1.4) | 10 (5.0) | 0.30 |
| Ticagrelor | 4 (1.5) | 2 (2.9) | 2 (1.0) | 0.27 |
| Warfarin | 164 (60.3) | 42 (60.0) | 122 (60.4) | 0.95 |
| Apixaban | 3 (1.1) | 0 | 3 (1.5) | 0.57 |
| Rivaroxaban | 7 (2.6) | 3 (4.3) | 4 (2.0) | 0.38 |
| **Laboratory results** | | | | |
| Hemoglobin, g/L, mean (SD) | 138.2 (15.5) | 129.4 (16.7) | 141.3 (16.7) | <0.01 |
| Creatinine, μmol/L, mean (SD) | 99.1 (36.3) | 87.2 (26.3) | 103.3 (38.4) | <0.01 |
| eGFR, mL/min/1.73m$^2$, mean (SD)# | 70.6 (22.7) | 64.2 (20.6) | 72.9 (23.1) | <0.01 |
| **HAS-BLED score** | | | | |
| HAS-BLED, median (25th, 75th percentiles)*** | 2.0 (2.0, 3.0) | 2.0 (2.0, 3.0) | 2.0 (1.0, 3.0) | 0.34 |

Figures presented as numbers (percentages) if not otherwise specified.

*Including gastro-, recto-, colo- or sigmoidoscopy

**No patients were treated with prasugrel, dabigatran or edoxaban.

#According to the MDRD, Modification of Diet in Renal Disease, equation. SD, standard deviation; TIA = Transient Ischemic Attack; PCI = Percutaneous Coronary Intervention; CABG = Coronary Artery Bypass Grafting; GI = gastrointestinal; NSAID = Non-Steroidal Anti-inflammatory Drugs; PPI = Proton Pump Inhibitors; eGFR, estimated Glomerular Filtration Rate;

***HAS-BLED score modified, including renal disease, liver disease, stroke history, prior major bleeding, age>65, medication usage predisposing to bleeding.

**Table 2. Interventions and events during index hospitalisation.**

|  | All (n = 272) | Women (n = 70) | Men (n = 202) | P-value |
|---|---|---|---|---|
| **Interventions** | | | | |
| Coronary angiography | 265 (97.4) | 67 (95.7) | 198 (98) | 0.38 |
| PCI | 254 (93.4) | 64 (91.4) | 190 (94.1) | 0.42 |
| Gastroscopy | 2 (0.7) | 0 (0.0) | 2 (1.0) | 1.00 |
| **Arterial access** | | | | |
| Radial | 161 (61.0) | 41 (62.1) | 120 (60.6) | 0.05 |
| Femoral | 97 (36.7) | 21 (31.8) | 76 (38.4) | |
| Alternate Access site* | 6 (2.3) | 4 (6.1) | 2 (1.0) | |
| **Bleeding during index hospitalisation** | | | | |
| Any bleeding | 68 (25.0) | 16 (22.9) | 52 (25.7) | 0.63 |
| Transfusion | 4 (1.5) | 3 (4.2) | 1 (0.5) | 0.02 |
| **Bleeding severity according to the TIMI classification** | | | | |
| Major | 1 (0.4) | 0 | 1 (0.5) | 0.90 |
| Minor | 11 (4.0) | 3 (4.3) | 8 (4.0) | |
| Minimal | 56 (20.6) | 13 (18.6) | 43 (21.3) | |
| **Bleeding severity according to the BARC classification** | | | | |
| Grade 3 | 3 (1.1) | 1 (1.4) | 2 (1.0) | 0.94 |
| Grade 2 | 23 (8.5) | 5 (7.1) | 18 (8.9) | |
| Grade 1 | 42 (15.4) | 10 (14.3) | 32 (15.8) | |
| **Localisation of bleeding** | | | | |
| Any gastrointestinal | 5 (1.8) | 2 (2.9) | 3 (1.5) | 0.36 |
| Any urogenital | 8 (2.9) | 1 (1.4) | 7 (3.5) | |
| Any ICA/PCI related bleeding | 38 (14.0) | 7 (10.0) | 31 (15.3) | |
| Any surgery related bleeding | 1 (0.4) | 1 (1.4) | 0 | |
| Other bleeding location# | 16 (5.9) | 11 (5.4) | 5 (7.1) | |

Figures presented as numbers (percentages) if not otherwise specified.

*Two interventions.

#Mostly epistaxis and hematomas. ICA, Invasive Coronary Angiography; PCI = Percutaneous Coronary Intervention; CABG = Coronary Artery Bypass Grafting; TIMI = Thrombolysis in Myocardial Infarction.

At the first bleeding event post-discharge 47.7% were treated with TAT, 44.0% were treated with aspirin and OAC, 1.8% were treated with DAPT and 0.9% were treated with $P2Y_{12}$-inhibitor and OAC. (Table 4).

Sixteen patients (5.9%) experienced any TIMI major bleeding and 61 (22.8%) any TIMI minor bleeding. According to BARC classification 1.5% experienced any BARC 4–5 bleeding and 9.6% any BARC 3 bleeding. Gastrointestinal bleeds (GIB) occurred in 13.6%, intracranial (IC) in 2.2%, urogenital in 5.1%, surgery related bleeds in 2.2% and bleeds related to coronary angiography and/or PCI in 3.3%. There were no sex differences in bleeding severity or bleeding location (Table 4).

In total 156 bleeds were registered, 109 in men and 47 in women. Among these, TIMI minor and BARC 2 were the most common (50.0 and 60.2%, respectively), and 46.8% were less serious bleeds such as epistaxis and hematomas, followed by GIB (28.8%), and urogenital (9.0%). Six bleeds (3.8%) were intracranial, of which five were in men.

Factors significantly associated with bleeding were age (per year increase, HR 1.03, 95% CI 1.00–1.05) creatinine on admission (per 1 μmol/L increase, HR 1.02, 95% CI 1.01–1.02) and warfarin initiated before index hospitalisation (HR 0.66, 95% CI 0.45–0.98). For all co-variables see S3 Table.

**Table 3. Therapy at discharge and premature discontinuation of TAT.**

| | All (n = 272) | Women (n = 70) | Men (n = 202) | P-value |
|---|---|---|---|---|
| **Type of antiplatelet and oral anticoagulant agents in TAT at discharge***  | | | | |
| Aspirin | 272 (100) | 70 (100) | 203 (100) | NA |
| Clopidogrel | 265 (97.4) | 68 (97.1) | 197 (97.5) | 1.00 |
| Ticagrelor | 7 (2.6) | 2 (2.9) | 5 (2.5) | 1.00 |
| Warfarin | 264 (97.1) | 67 (95.7) | 197 (97.5) | 0.43 |
| Apixaban | 6 (2.2) | 2 (2.9) | 4 (2.0) | 0.65 |
| Rivaroxaban | 2 (0.7) | 1 (1.4) | 1 (0.5) | 0.45 |
| **Other medication at discharge** | | | | |
| NSAID | 0 | 0 | 0 | NA |
| PPI | 138 (50.7) | 44 (62.9) | 94 (46.5) | 0.02 |
| **Planned duration of TAT** | | | | |
| ≤1 month | 168 (61.8) | 47 (67.1) | 121 (59.9) | 0.22 |
| 1–3 months | 75 (27.6) | 17 (24.3) | 58 (28.7) | |
| 3–6 months | 27 (9.9) | 5 (7.1) | 22 (10.9) | |
| 6–12 months | 2 (0.7) | 1 (1.4) | 1 (0.5) | |
| **Discontinuation or interruption of TAT#** | | | | |
| Any discontinuation | 110 (40.4) | 37 (52.9) | 73 (36.1) | 0.01 |
| Aspirin discontinuation | 44 (16.2) | 18 (25.7) | 26 (12.9) | 0.01 |
| P2Y$_{12}$-inhibitor discontinuation | 31 (11.4) | 12 (17.1) | 19 (9.4) | 0.08 |
| OAC discontinuation | 91 (33.5) | 26 (37.1) | 65 (32.2) | 0.45 |
| **Reason for discontinuation or interruption of TAT** | | | | |
| Gastric symptoms | 3 (2.7) | 3 (8.1) | 0 | 0.01 |
| Bleeding | 49 (44.5) | 17 (45.9) | 32 (43.8) | 0.83 |
| Allergic reaction | 2 (1.8) | 0 | 2 (2.7) | 0.55 |
| Coronary angiography/PCI | 20 (18.2) | 5 (13.5) | 15 (20.5) | 0.37 |
| Surgery | 20 (18.2) | 5 (13.5) | 15 (20.5) | 0.37 |

Figures presented as numbers (percentages) if not otherwise specified.

*No patient received prasugrel, dabigatran or edoxaban at discharge.

#Among patients that interrupted or discontinued TAT. TAT = Triple Antithrombotic Therapy; OAC = Oral Anticoagulant; GI = Gastrointestinal; PCI = Percutaneous Coronary Intervention; CABG = Coronary Artery Bypass Grafting; TIMI = Thrombolysis in Myocardial Infarction; BARC = Bleeding Academic Research Consortium.

*Two interventions.

## Mortality during follow-up

Twenty patients died during follow-up, 11.4% women and 5.9% men. (p = 0.13). (Table 4) Thirteen of these patients had suffered at least one bleeding complication during or after index hospital discharge.

## Costs of bleeding events

The per patient mean health care costs were EUR 5787 and EUR 575 for bleeders and non-bleeders respectively during the follow-up period. Health care costs were similar for men and women with and without bleeds during follow-up (Table 5).

## Discussion

In total 54.4% (58.6% women, 53.0% men, p = 0.41) experienced a bleeding during or within one year after the index event. The most important finding in the present study was the

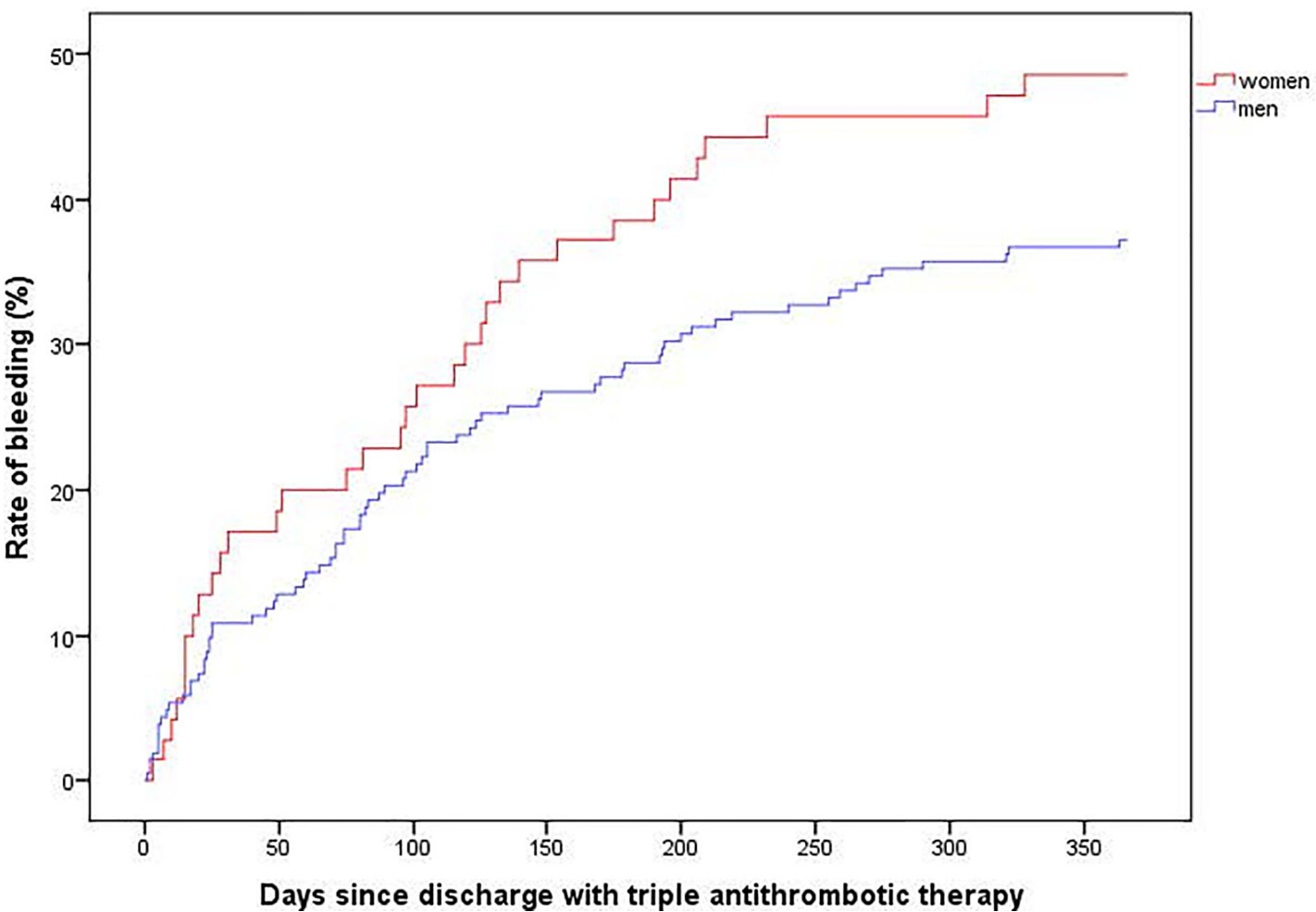

**Fig 1. Long term risk of bleeding in patients discharged with triple antithrombotic therapy.** Log rank test men vs women 0.09.

extremely high incidence of more than 40% of patients bleeding within one year after discharge and a cumulative overall incidence rate of over 50%. The 272 patients suffered from 156 bleeds post-discharge, and from 224 bleeds if the index hospitalisation was included. Women had almost 50% higher risk to bleed than men, not reaching statistical significance (HR 1.47, 95% CI 0.94–2.31, p = 0.09). Women discontinued TAT prematurely due to bleeds to a very high extent compared to men.

Previous studies have shown a two to three-fold higher risk of bleeding with TAT compared to DAPT [16,23,24]. TAT has also been found associated with a higher risk of bleeding compared to OAC plus one antiplatelet drug, so called dual antithrombotic therapy (DAT) [25]. In another Swedish cohort 10.2% of TAT-treated patients experienced a major bleeding within one year [24,26]. This is concurrent to 6% having a TIMI major bleeding and 8% a BARC 3–5 bleeding in the current study. A Danish registry study found a three year incidence of 15.7% of bleeds leading to hospitalisation or death in TAT treated AF patients [17]. In contrast to previous studies in this area, we included all bleeds receiving medical attention and all indications for TAT and found more than double the rate compared to others. Almost one fourth of the patients had a TIMI minor bleeding, which is potentially dangerous and even more had a BARC 2 bleeding, demanding hospitalisation and/or intervention. All bleeds leading to

**Table 4. Bleeding and ischemic events during follow-up.**

| | All (n = 272) | Women (n = 70) | Men (n = 202) | P-value |
|---|---|---|---|---|
| **Bleeding during follow-up** | | | | |
| Any bleeding | 109 (40.1) | 34 (48.6) | 75 (37.1) | 0.09 |
| **Number of bleeds during follow-up** | | | | |
| One | 78 (28.7) | 24 (34.3) | 54 (26.7) | 0.38 |
| Two | 18 (6.6) | 7 (10.0) | 11 (5.4) | |
| Three | 10 (3.7) | 3 (4.3) | 7 (3.5) | |
| Four | 3 (1.1) | 0 | 3 (1.5) | |
| **Therapy at first bleeding event** | | | | |
| Triple Antithrombotic Therapy | 52 (47.7) | 15 (44.1) | 37 (49.3) | 0.45 |
| Dual Antiplatelet Therapy | 2 (1.8) | 1 (2.9) | 1 (1.3) | |
| Aspirin and OAC | 48 (44.0) | 17 (50.0) | 31 (41.3) | |
| P2Y$_{12}$-inhibitor and OAC | 1 (0.9) | 1 (2.9) | 0 | |
| Aspirin, single therapy | 2 (1.8) | 0 | 2 (2.7) | |
| P2Y$_{12}$-inhibitor, single therapy | 0 | 0 | 0 | |
| OAC, single therapy | 1 (0.9) | 0 | 1 (1.3) | |
| **Any bleeding during follow-up according to the TIMI classification** | | | | |
| Major | 16 (5.9) | 3 (4.3) | 13 (6.4) | 0.51 |
| Minor | 62 (22.8) | 19 (27.1) | 43 (21.3) | 0.31 |
| Minimal | 46 (16.9) | 16 (22.9) | 30 (14.9) | 0.12 |
| **Any bleeding during follow-up according to the BARC classification** | | | | |
| BARC 5 | 3 (1.1) | 0 | 3 (1.5) | 0.57 |
| BARC 4 | 1 (0.4) | 0 | 1 (0.5) | 0.56 |
| BARC 3 | 26 (9.6) | 7 (10.0) | 19 (9.4) | 0.88 |
| BARC 2 | 72 (26.5) | 23 (32.9) | 49 (24.3) | 0.16 |
| BARC 1 | 24 (8.8) | 10 (14.3) | 14 (6.9) | 0.06 |
| **Localisation of bleeding** Any bleeding with the following localisation; | | | | |
| Intracranial | 6 (2.2) | 1 (1.4) | 5 (2.5) | 1.00 |
| Gastrointestinal | 37 (13.6) | 14 (20.0) | 23 (11.4) | 0.07 |
| Urogenital | 14 (5.1) | 1 (1.4) | 13 (6.4) | 0.13 |
| ICA/PCI related | 9 (3.3) | 2 (2.9) | 7 (3.5) | 1.00 |
| Surgery related | 6 (2.2) | 0 | 6 (3.0) | 0.34 |
| Other location* | 60 (22.1) | 20 (30.0) | 40 (19.3) | 0.06 |
| **Ischemic Events during follow-up** | | | | |
| Any coronary event | 49 (18.0) | 13 (18.6) | 36 (17.8) | 0.89 |
| Stable CAD | 23 (8.5) | 7 (10.0) | 16 (7.9) | 0.59 |
| UAP | 20 (7.4) | 4 (5,7) | 16 (7.9) | 0.54 |
| NSTEMI | 12 (4.4) | 4 (5.7) | 12 (4.4) | 0.54 |
| STEMI | 2 (0.7) | 1 (1.4) | 1 (0.5) | 0.45 |
| PCI | 15 (5.5) | 3 (4.3) | 12 (5.9) | 0.65 |
| stroke/TIA | 8 (2.9) | 2 (2.9) | 6 (3.0) | 1.00 |
| **Mortality** | | | | |
| Dead during follow-up | 20 (7.4) | 8 (11.4) | 12 (5.9) | 0.13 |

Figures presented as numbers (percentages) if not otherwise specified.

*Mostly epistaxis or hematomas. OAC = Oral Anticoagulant; TIMI = Thrombolysis in Myocardial Infarction; BARC = Bleeding Academic Research Consortium; SCAD = Stable Coronary Artery Disease; UAP = Unstable Angina Pectoris; NSTEMI = Non-ST-Elevation Myocardial Infarction; STEMI = ST-Elevation Myocardial Infarction; PCI = Percutaneous Coronary Intervention.

**Table 5. Unit costs and one-year health care costs in bleeders and non-bleeders, EUR, Euro.**

| Resource use | Unit costs | No bleeding n = 162 | | Bleeding n = 110 | | Difference (bleeders vs not bleeders) |
|---|---|---|---|---|---|---|
| | | Mean | SEM | Mean | SEM | |
| Bleeding defined | | | | | | |
| Hospitalization due to bleeding | 376; 547* | 0 | 0 | 2893 | 518 | |
| Bleeding during hospitalization** | 547*** | 0 | 0 | 56 | 21 | |
| Outpatient care visit due to bleeding | 312 | 0 | 0 | 192 | 21 | 192 |
| Blood products | 101 | 0 | 0 | 104 | 21 | 104 |
| *Cost bleeding defined* | | *0* | *0* | *3236* | *517* | *3236* |
| Costs Not necessarily due to bleedings**** | | | | | | |
| Hospitalization due to ACS | 376; 547* | 314 | 80 | 1934 | 638 | 1620 |
| Sigmoidoscopy | 456 | 0 | 0 | 17 | 8 | 17 |
| Rectoscopy | 456 | 6 | 4 | 80 | 19 | 74 |
| Colonoscopy | 587 | 7 | 5 | 38 | 14 | 31 |
| Gastroscopy | 456 | 8 | 5 | 126 | 23 | 117 |
| CABG | 6476 | 80 | 56 | 118 | 84 | 39 |
| PCI | 3255 | 160 | 55 | 239 | 82 | 79 |
| *Cost not necessarily due to bleeding* | | *575* | *113* | *2553* | *721* | *1978* |
| Total costs | | 575 | 113 | 5790 | 928 | 5215 |

SEM, Standard Error of the Mean.

*Hospitalizations are costed by 376 EUR plus 547 EUR per bed day.

**Bleeding related cost during hospitalization for other reason than bleeding or ACS.

***Per bed day associated with the bleeding.

****Of which some may be related to bleeding but not collected solely as a bleeding event.

medical attention may not be life-threatening but lead to suffering for the patients and costs for the society. In the TRANSLATE-ACS study on 9000 DAPT treated patients all severity grades of bleeds were associated with lower quality of life, even BARC type 1 [27].

Although the design and focus of the current study was not on resource use and costs, the extensive data collection still allowed us to explore health care costs associated with bleeding. The results indicate that bleeding is not only associated with patient suffering but also with extensive health care costs. One-year per patient mean health care costs for patients with at least one bleeding episode were EUR 5787 during follow-up, of which EUR 3236 with certainty were associated with bleeding-related resource use. Of importance, the higher cost in bleeders vs non-bleeders was not only due to the bleeds themselves, but also to concomitant ischemic events including PCI or CABG (36.4% stopped TAT due to PCI or surgery). Thus, the bleeding was often a consequence of, and not a reason to, the revascularisation event.

Bleeding complications could differ substantially in seriousness and therefore in healthcare costs as well as subsequent impact on quality of life for the individual. For instance, in the bleeding group six patients suffered from intracranial haemorrhage, which may lead to death or disability. Such catastrophic bleeding event is of course far more important than nose bleeds, hematomas and even GIB, with long-term economic burden for patients, their families, the healthcare and other societal systems.

The most common bleeding site was the gastrointestinal tract. One fifth of all women compared to one tenth of all men developed GIB and among the 156 bleeds more than one forth were GIB. These bleeds could be dramatic and life-threatening such as in profound bleeding

from peptic ulcers, but also slow and undiscovered for a long time such as bloody stools [28]. TAT increases the risk of GIB 3 to 6-fold compared to each drug that constitutes TAT alone, or DAT [28]. Bleeders in our study used PPI more often than non-bleeders before admission, probably identifying a high risk group for bleeding, consistent with studies showing that any GIB is a predictor for a new GIB event [28]. Thus, PPI use should be taken into account when prescribing TAT, as an indicator of an increased risk of bleeding. In addition, patients with TAT should be prescribed PPI protecting them from GIB [29,30], according to current guidelines [13,31]. This was only done in half of the patients in the present study.

It is known that women discontinue with DAPT more often than men [32,33], and that they have a higher risk of bleeding. [9,11,34–36]. In the current study women tolerated TAT less well than men and discontinued TAT to a high extent, most often due to bleeds. This may represent a true sex difference that we could not discover because of the limited size of the study cohort. A surprisingly high rate of both men and women prescribed TAT had bled before discharge–in several cases even serious bleeds. In 43.5% of the patients that had a bleeding event during the index hospitalisation a new bleeding occurred during follow-up. Thus, a previous bleed ought to lead to an increased attention and in most cases a more cautious therapy with DAT. In addition, eighteen patients (6.6%) were treated with TAT although PCI was not performed. In these patients DAT is a better choice in order to avoid an increased bleeding risk, and is the recommended treatment [13,37].

The rationale behind TAT is to reduce the risk of coronary ischemic as well as thromboembolic events, and that neither DAPT nor OAC is sufficient to prevent complications [31]. Especially, the fear of stent thrombosis (ST) post PCI has prevailed the use of DAPT also in OAC treated patients [38,39]. According to the recently performed meta-analysis on patients with an indication for TAT, no difference between DAT and TAT treated groups was found in major adverse cardiac events (10.4% vs. 10.0%, HR 0.85, 95% CI 0.48–1.29) [40].

The most common indication for OAC in the present study was AF (71.3%). During the recruitment time (2009–2015) the guidelines recommended TAT for a certain time period after PCI in AF patients, and warfarin was the most commonly used OAC. Our results show that more than half of the patients treated with this regime bled. During the years after our recruitment period, several studies have been published showing that DAT is preferable to TAT in this context [23,41–43]. In addition, DOAC has been found less harmful than warfarin, as shown in the AUGUSTUS trial, where apixaban was compared with warfarin, on top of a $P2Y_{12}$-inhibitor [44]. Unfortunately, neither of these trials was adequately powered to properly address the question whether any combination of DAT is enough to protect from thromboembolic and coronary ischemic events, including ST. In spite of these results, TAT including warfarin is still widely used in Europe [45]. Today, DOAC is preferred over warfarin in international guidelines on AF with ACS events [31,46]. Anyhow, the European and American guidelines are discordant whether TAT or DAT should be the first line choice for the majority of patients after hospital discharge. While a limited period of TAT with clopidogrel as the $P2Y_{12}$-inhibition of choice is recommended in the European guidelines [13,37], the American guidelines recommends DAT as first line therapy, including DOAC plus a choice between ticagrelor and clopidogrel as $P2Y_{12}$-inhibition [46].

One of our aims was to study sex differences regarding bleeding and discontinuation of TAT. We found a high bleeding risk in both sexes, although the bleeding risk according to HAS-BLED was intermediate. There was a trend, but not statistically significant, to more bleedings in women and discontinuation of TAT was significantly more common. Despite this, the rate of coronary events did not differ between sexes. Anyhow the current study is underpowered to assess a possible sex difference in association between bleeding/TAT discontinuation and ischemic events. To our knowledge there are no gender analyses published from

the big RCTs regarding DAT and TAT to guide in treatment strategies of ACS-patients in need of both anticoagulation and platelet inhibition [44].

To us it seems reasonable, as also stated by Capodanno et al in a recent State-of-the Art Review on AF [47], to risk-identify each individual patient regarding both bleeding and ischemic events and treat accordingly. Thus, if indication for TAT for any reason, treat as short as possible (usually one month) or not at all using DAT instead. In addition, as many bleeding-avoidance strategies as possible should be used, such as radial access and PPI at discharge.

## Conclusion

In summary, TAT confers a high risk of bleeding complications. It is important to consider the ischemic risk profile as well as predictors of bleeding complications, to be able to assess each patient properly and individualise treatment. There is a need to implement cost-effective strategies in this population, taking into consideration the risk of new bleeds as well as ischemic events. International guidelines are still discordant on how to best treat this patient population. Thus, there is a need of a randomised controlled trial on all indications for TAT, adequately powered to assess coronary ischemic and thromboembolic risks, comparing TAT with DAT including DOAC in combination with a potent $P2Y_{12}$-inhibitor.

## Strengths and limitations

This study was designed as a retrospective, medical record-based analysis with the limitations inherent to such study designs. We anticipated 10% of the ACS population having an indication for OAC, but only half of these were prescribed TAT at discharge. The current study was performed in an era of warfarin treatment while nowadays DOACs are more commonly used and this could limit the generalisability of the study. Anyhow, even today warfarin is still widely used in Europe [46], In addition, in the current study population almost one third of patients discharged with TAT had another indication than AF, such as mechanical valves, and were treated with warfarin accordingly.

The study population size probably limited the possibility to reach statistical significance comparing the sexes. Moreover, the present study lacked information about how many were not discharged with TAT despite indication. Finally, the costs are not complete in this study, as we did not account for indirect costs e.g. sick leave, disabilities and caretaker obligations.

The strengths of the study are the thorough examination of patient files in order to register all forms of bleeds leading to medical attention, as well as the broad inclusion of patients receiving TAT, not restricting the cohort to AF patients undergoing PCI. Another strength is the long-time follow-up.

## Supporting information

**S1 Table. Subgroup analysis of patients with atrial fibrillation.**
(DOCX)

**S2 Table. Bleeding events related to planned TAT duration.**
(DOCX)

**S3 Table. Cox regression model, time to bleeding.** All co-variates displayed.
(DOCX)

## Acknowledgments

### Authors' relationships and activities

A.C. Holm, S Sederholm Lawesson and E Swahn are responsible for the conception and design of the study, have full access to all data, analysed and interpreted the data and drafted the manuscript. M Henriksson analysed and drafted the health economic part of the study and revised the manuscript from a health economists´ view. T Johansson and D Vial collected data from patient files. J Alfredsson and M Janzon critically revised the manuscript and added some important intellectual content.

### Patient and public involvement

Patients or the public were not involved in the design, or conduct, or reporting, or dissemination plans of our research.

## Author Contributions

**Conceptualization:** Anna Holm, Eva Swahn, Sofia Sederholm Lawesson.

**Data curation:** Anna Holm, Therese Johansson, Eva Swahn, Dominique Vial, Sofia Sederholm Lawesson.

**Formal analysis:** Anna Holm, Martin Henriksson, Eva Swahn, Sofia Sederholm Lawesson.

**Funding acquisition:** Eva Swahn.

**Investigation:** Anna Holm, Eva Swahn, Sofia Sederholm Lawesson.

**Methodology:** Anna Holm, Martin Henriksson, Eva Swahn, Sofia Sederholm Lawesson.

**Project administration:** Eva Swahn.

**Resources:** Eva Swahn.

**Software:** Eva Swahn.

**Supervision:** Eva Swahn, Sofia Sederholm Lawesson.

**Validation:** Anna Holm, Martin Henriksson, Joakim Alfredsson, Magnus Janzon, Eva Swahn, Sofia Sederholm Lawesson.

**Visualization:** Anna Holm, Sofia Sederholm Lawesson.

**Writing – original draft:** Anna Holm, Martin Henriksson, Eva Swahn, Sofia Sederholm Lawesson.

**Writing – review & editing:** Joakim Alfredsson, Magnus Janzon, Therese Johansson, Eva Swahn, Dominique Vial.

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
