## [Decision Letter · Decision Letter 0]

15 Dec 2020

PONE-D-20-34276

Long term risk and costs of bleeding in men and women treated with triple antithrombotic therapy – an observational study

PLOS ONE

Dear Dr. Swahn,

Thank you for submitting your manuscript to PLOS ONE. After careful consideration, we feel that it has merit but does not fully meet PLOS ONE’s publication criteria as it currently stands. Therefore, we invite you to submit a revised version of the manuscript that addresses the points raised during the review process, especially those raised by reviewer #2

We look forward to receiving your revised manuscript.

Kind regards,

Raffaele Bugiardini, M.D.

Academic Editor

PLOS ONE

Journal Requirements:

Reviewers' comments:

Reviewer's Responses to Questions

**Comments to the Author**

1. Is the manuscript technically sound, and do the data support the conclusions?

Reviewer #1: Yes

Reviewer #2: Partly

2. Has the statistical analysis been performed appropriately and rigorously? 

Reviewer #1: Yes

Reviewer #2: Yes

3. Have the authors made all data underlying the findings in their manuscript fully available?

Reviewer #1: Yes

Reviewer #2: Yes

4. Is the manuscript presented in an intelligible fashion and written in standard English?

Reviewer #1: Yes

Reviewer #2: Yes

5. Review Comments to the Author

Reviewer #1: Dear Author,

This is a well-designed retrospective study that uses clinical records as a source of data.

The topic of gender difference in presentation, and the clinical course and differences in response to therapeutic treatments in patients with acute coronary heart disease between men and women has been particularly topical in recent years.

From the previous studies we know that bleeding complications are associated more frequently with female gender.

In this study authors are reporting unusually high rate of bleeding complications in patients with TAT in both genders, however they underline the significance of potential under power of the study, the small number of subjects that potentially influence the final results.

Findings in the study are close to the data reported in the literature: women tolerated TAT less well than men and discontinued TAT to a high extent, most often due to bleeds. This may represent a true sex difference that they could not discover because of the limited size of the study cohort.

The reference list is comprehensive with respective to number of references, time period and coverage of the questions of interest.

COMMENT:

Quote: “Most patients (61.8 %) were planned for ≤1 month of TAT followed by 11 months of aspirin and OAC combined, followed by OAC only.”

This is a surprising report, as according to the Guidelines when TAT stops OAK and Clopidogrel is recommended combination. This is interesting from the point of treatment strategy in the country of origin of this paper.

Minor remarks with respect to grammar and typos:

1. Page 5

Baseline characteristics subheading

.On admission 60.3% was (were) treated with OAC

2. Legend for Table 1

Typo: endoxaban should be edoxaban

Reviewer #2: Holm et al. report their findings on sex-differences in one-year bleeding rates in patients receiving triple antiplatelet therapy (TAT) as well as related health care costs. Data are drawn during a 7-year span from 3 centers of the SWEDEHEART registry. Bleeding was defined according to the TIMI and BARC definitions. Resource use data retrieved in the study included hospitalizations, outpatient care visits, procedures and blood products. Among the study population (25.7% women), 54.4% had at least one bleeding event during or after the index event and 40.1% bled post discharge of whom 28.7% experienced a TIMI major or minor bleeding. GI bleedings occurred in 28.8% of patients’ post-discharge. Women were more likely than men to experience bleeding events and to discontinue TAT prematurely. One-year mean health care costs were approximately 10 times higher in patients who experience bleeding complications.

Comments:

- Abstract: the authors should state the background of the study in order provide the context of their research

- Full adjusted cox model of factors associated with bleeding should be reported and displayed

- Reductions in bleeding complications have become a primary target for further improvements in both clinical and economic outcomes. Nevertheless, knowledge about the relative size of the costs of bleeding events compared to coronary ischemic, ischemic stroke and/or VTE events in patients treated with TAT is a valuable evidence for decision makers on risk benefits. E.g as shown in table 5 PCI or CABG cost far exceed those of bleeding. Of utmost importance the majority of patients had minor or GI bleeding (e.g. DOACs therapy is more frequently associated with GI bleedings) which are manageable in terms of clinical outcomes as compared with intracranial hemorrhage. The later in addition to an ACS/AMI, ischemic stroke may lead to death or disability which in turn are associated with impaired quality of life and an important chronic and long-term economic burden for both patients/families and health systems. Therefore, the authors should discuss these aspects in the discussion section and limitation in order to give to the readers knowledge about the cost of bleeding relative to those of major events to improve decision making

- Again, in the discussion section there are too many editor comments. The authors should focus more on discussion their findings

- The reviewer urges the authors to temper their conclusions. It is not new that TAT is associated with higher bleeding risk and recent guidelines recommend a shorter time of TAT therapy. Nevertheless, the reviewer agrees with the authors that bleeding complications following antithrombotic therapy is associated with higher costs and that there is a need to implement cost-effective strategies in these population

6. PLOS authors have the option to publish the peer review history of their article (what does this mean?). If published, this will include your full peer review and any attached files.

Reviewer #1: No

Reviewer #2: No

---

## [Author Response · Author response to Decision Letter 0]

19 Feb 2021

Respons to reviewers PLOS ONE 17th of Feb 2021

PONE-D-20-34276

Long term risk and costs of bleeding in men and women treated with triple antithrombotic therapy – an observational study 

Reviewer #1

Comment R1 No 1: Quote: “Most patients (61.8 %) were planned for ≤1 month of TAT followed by 11 months of aspirin and OAC combined, followed by OAC only.” This is a surprising report, as according to the Guidelines when TAT stops OAK and Clopidogrel is recommended combination. This is interesting from the point of treatment strategy in the country of origin of this paper.

Answer R1 No 1: Thank you for this comment. This was actually the guideline recommendation at that time – i.e., to stop either aspirin or P2Y12-inhibition when TAT was stopped. After one year of dual treatment, only OAC continued (this is still the case according to current guidelines).

Comment R1 No 2: Page 5 Baseline characteristics subheading. On admission 60.3% was (were) treated with OAC.

Answer R1 No 2: Thank you, this typo is now corrected. 

Comment R1 No 3: Legend for Table 1. Typo: endoxaban should be edoxaban.

Answer R1 No 3: Thank you, this typo is now corrected. 

Reviewer #2

Comments R2 No 1: Abstract: the authors should state the background of the study in order provide the context of their research.

Answer R1 No 1: Thank you for this comment. The abstract is now updated accordingly. 

Comments R2 No 2: Full adjusted cox model of factors associated with bleeding should be reported and displayed.

Answer R2 No 3: Thank you for this comment. The full model is now added as a supplementary file. 

Comments R2 No 4: Reductions in bleeding complications have become a primary target for further improvements in both clinical and economic outcomes. Nevertheless, knowledge about the relative size of the costs of bleeding events compared to coronary ischemic, ischemic stroke and/or VTE events in patients treated with TAT is a valuable evidence for decision makers on risk benefits. E.g as shown in table 5 PCI or CABG cost far exceed those of bleeding. Of utmost importance the majority of patients had minor or GI bleeding (e.g. DOACs therapy is more frequently associated with GI bleedings) which are manageable in terms of clinical outcomes as compared with intracranial hemorrhage. The later in addition to an ACS/AMI, ischemic stroke may lead to death or disability which in turn are associated with impaired quality of life and an important chronic and long-term economic burden for both patients/families and health systems. Therefore, the authors should discuss these aspects in the discussion section and limitation in order to give to the readers knowledge about the cost of bleeding relative to those of major events to improve decision making.

Answer R2 No 4: Thank you for this very valuable comment. We have now expanded the discussion, and we think that your input has helped us to strengthen the discussion substantially. We agree that it is a limitation of the study that we do not have complete information on all costs – including indirect costs. We have added this information into the limitation section as well.

Comment R2 No 5: Again, in the discussion section there are too many editor comments. The authors should focus more on discussion their findings.

Answer R2 No 5: Thank you for the comment. We have now changed the discussion section substantially and focused more on our own results.

Comments R2 No 6: The reviewer urges the authors to temper their conclusions. It is not new that TAT is associated with higher bleeding risk and recent guidelines recommend a shorter time of TAT therapy. Nevertheless, the reviewer agrees with the authors that bleeding complications following antithrombotic therapy is associated with higher costs and that there is a need to implement cost-effective strategies in these population

Answer R2 No 6: Thank you for the comment. We have now tempered our conclusions.

---

## [Editor Report · Decision Letter 1]

25 Feb 2021

Long term risk and costs of bleeding in men and women treated with triple antithrombotic therapy – an observational study

PONE-D-20-34276R1

Dear Dr. Swahn,

We’re pleased to inform you that your manuscript has been judged scientifically suitable for publication and will be formally accepted for publication once it meets all outstanding technical requirements.

Kind regards,

Raffaele Bugiardini, M.D.

Academic Editor

PLOS ONE

---

## [Editor Report · Acceptance letter]

8 Mar 2021

PONE-D-20-34276R1 

Long term risk and costs of bleeding in men and women treated with triple antithrombotic therapy – an observational study 

Dear Dr. Swahn:

I'm pleased to inform you that your manuscript has been deemed suitable for publication in PLOS ONE. Congratulations! Your manuscript is now with our production department. 

Kind regards, 

on behalf of

Prof. Raffaele Bugiardini 

Academic Editor

PLOS ONE